# Knowledge, attitudes, and practices of healthcare professionals regarding rabies in tertiary care hospitals: A cross-sectional study in Peshawar, Pakistan

Adnan Ahmad[1], Fawad Inayat[2], Noor Ullah[3], Shaista Rasul[4], Shehnaz Bakhtiar[5]*, Zar Shad[6], Zakir Ahmad[2]

1 Department of Allied Health Sciences, Cecos University of IT and Emerging Sciences Peshawar, Pakistan, 2 Department of Biochemistry, Institute of Chemical and Life Science, Abdul Wali Khan University Mardan, Pakistan, 3 Institute of Paramedical Sciences, Khyber Medical University, Peshawar, Pakistan, 4 Institute of Public Health and Social Sciences, Khyber Medical University Peshawar, Peshawar, Pakistan, 5 Khyber Institute of Child Health and Bashir Bilour Memorial Children Hospital, Peshawar, Pakistan, 6 Department of Biological and Health Sciences, Abasyn University, Peshawar, Pakistan

* Bakhtiarshehnaz@gmail.com

**Data Availability Statement:** All relevant data are in the manuscript and its supporting information files.

## Abstract

### Background

Rabies, caused by the rhabdovirus, is a fatal zoonotic disease with over 59,000 annual deaths globally. Asia and Africa account for 95%, with India leading, followed by China. In Pakistan, where it's endemic, over 50,000 cases are reported yearly. Controlling rabid dog populations through vaccination is crucial in curbing mortality. This research aimed to evaluate healthcare professionals' knowledge, attitudes, and practices concerning rabies in Peshawar, Pakistan.

### Methods

The study was conducted at different tertiary care hospitals in Peshawar, Pakistan from 16 August 2021 to 15 February 2022. Cross-sectional research was conducted to gather data from a total of 100 healthcare workers representing different sections within the healthcare field, including Medical Officers, House Officers, Faculty Staff, Nurses, and Paramedics. Data on knowledge, attitudes, and practices about rabies were collected using a standardized questionnaire. The data analysis included using descriptive statistics and chi-square testing to ascertain potential correlations.

### Results

Among the healthcare professionals, 68 (68%) were males, and 32 (32%) were females. Profession-wise, the included professionals were Nurses 31 (31%), Medical Officers 27 (27%), House officers 26 (26%), paramedical staff 13 (13%), and faculty staff 3 (3%). 91 (91%) and 9 (9%) healthcare professionals responded that dogs and cats are responsible for rabies transmission, respectively. Moreover, 82 (82%) individuals responded that animal

**Funding:** The author(s) received no specific funding for this work.

**Competing interests:** The authors have declared that no competing interests exist.

bite plays a vital role in the transmission of rabies, whilst 76 (76%) individuals responded that rabies transferred from human to human. 82 (82%) individuals replied that the anti-rabies vaccine (ARV) is the treatment of choice for rabies. Furthermore, 78 (78%) individuals responded that ARV is safe in pregnancy and lactation. Moreover, after being asked about the perception of the health care professionals about the failure in controlling rabies, their responses were unavailability of ARV/RIG 41 (41%), lack of control of stray dogs 34 (34%), lack of awareness 20 (20%). The study revealed statistically significant correlations between healthcare occupations and variables: knowledge of animals responsible for transmitting rabies (p = 0.024) and awareness of human-to-human transmission (p = 0.007). Significant disparities were noted in understanding rabies transmission through contaminated water (p = 0.002). There were variations in attitudes and practices seen across different positions, particularly about views about home treatments (p = 0.033) and the perceived effectiveness of cleansing bite wounds (p = 0.010). Disparities in perceptions of rabies treatment and the accessibility of anti-rabies vaccines and immunoglobulin were observed, with variations based on individual roles.

## Conclusion

The present research elucidates variations in rabies knowledge, attitudes, and practices among healthcare workers, specifically concerning their respective roles. Tailored training programs and standardized practices play a crucial role in mitigating these discrepancies, fostering a greater understanding of rabies, and enhancing the quality of patient treatment. It is recommended that future studies undertake an assessment of the efficacy of therapies and advocate for the adoption of collaborative One Health strategies in the realm of rabies management.

## Author summary

This study examines healthcare professionals' knowledge, attitudes, and practices regarding rabies in Peshawar, Pakistan. There are disparities among professions in terms of awareness levels regarding transmission, treatment, and prevention. Nurses and medical officers display a higher level of awareness, while house officers and paramedics demonstrate a deficit in certain aspects. It is recommended to implement tailored training programs to address these gaps and improve the quality of patient care. The study emphasizes the importance of access to post-exposure prophylaxis and public awareness. Future research should focus on intervention effectiveness and collaborative One Health strategies. Despite limitations, this study contributes significantly to rabies control efforts, potentially saving lives and improving public health outcomes.

## 1 Introduction

The drastic paralysis of the central nervous system is caused by a fatal zoonotic disease called rabies. It is a widespread infectious disease considered the oldest and substantially affects all mammals [1,2]. Rabdovirus, a causative disease agent, is transmitted from rabid animals to humans via animal bite [3]. Annually, 59,000 mortalities or more are associated with rabies,

which ascribes it to position 11 in deadly infectious diseases globally [4,5]. Comparatively, the mortality rate is high in Asia and Africa (95% of total deaths) [5–7].

The subcontinent accounts for 30,000 rabies-associated deaths, followed by Africa [4]. Country Wise, the high mortalities due to rabies occur in India, followed by China [8]. Furthermore, rabies is endemic in Pakistan due to a lack of prevention and control programs [3,4]. Moreover, being endemic in Pakistan, more than 50,000 cases are reported annually, with over 6000 people losing their worthwhile lives [2,4].

Being affected by rabies, the incidence density of death ranges from 20 to 30 cases per million per annum in India. In contrast, the annual incidence rate of rabies deaths is 14 cases per million in Bangladesh, followed by Pakistan (7 to 9 cases) [9–11]. It's well known that dogs play a significant role in rabies transmission. Furthermore, the high mortality in Pakistan, India, and China is due to rabid dog bites. Moreover, the critical role of reducing the ad of rabies is controlling the rabid dog population and vaccination of dogs [5,12].

Basic knowledge, personal attitude, and related practices toward the disease and animals are crucial for rabies prevention and control. In rabies, endemic areas of the United States, Srilanka, Tanzania, Nigeria, considerable knowledge, attitude, and practice (KAP) assessments have been performed and reported a rabies outbreak in wildlife [13–17]. These surveys reveal people's perception of rabies, but additional information may be insubstantial regarding the infection. The KAP survey will identify knowledge gaps concerning rabies infection. Additionally, it will scrutinise the behaviour and cultural beliefs that possibly pretend to be a barrier to preventing infectious disease, i.e., zoonotic disease [13,18].

Likewise, the KAP study is helpful in the public health awareness campaign that provides the facilities of baseline data for planning, application, and evaluation of the national disease control programme [13].

This survey aimed to evaluate the rabies KAP status of health care professionals at different tertiary care hospitals in Pakistan. To the best of our knowledge, no detailed study was conducted to elaborate the health care personal association with rabies infection in clinical setup. The current study's findings may be helpful to plan future rabies control strategies training for health care professionals in Pakistan, providing valuable insights for improving rabies awareness and management in the healthcare sector.

## 2 Materials and methods

### Ethical approval

The study was approved by Advanced Study and Research Board (ASRB), Institute of Public Health and Social Sciences (IPH&SS), Khyber Medical University, Peshawar, Pakistan (DIR/ KMU-EB/MS/00073).

### Study design

This cross-sectional enrolled different healthcare professionals.

### Study settings

This study was conducted in tertiary care hospitals of Peshawar, they are Lady Reading Hospital (LRH), Khyber Teaching Hospital (KTH), and Hayatabad Medical Hospital (HMC). The three hospitals were selected for the following three reasons.

- Reference Centers for Rabies Management

- High Patient Influx

- Comprehensive Rabies Treatment Protocols

### Roles of professional categories in rabies patient care

Medical Officers have the main responsibility of doing the first evaluation and diagnosis of patients who have been injured by animals that have the potential to transmit rabies. The decision to provide rabies prophylaxis is made by evaluating the kind of exposure and the clinical symptoms. The nurses have a vital role in giving post-exposure prophylaxis, delivering wound care, and overseeing patients for any unfavorable responses to rabies immunoglobulin and immunizations, guaranteeing prompt treatment of such occurrences. House Officers provide support in the provision of emergency medical care and treatment to patients. They have the duty of promptly addressing wounds, evaluating the likelihood of rabies exposure, and commencing preventive therapy while being overseen by Medical Officers. Paramedics play a vital role in the pre-hospital environment by evaluating and treating patients at the location of an occurrence involving animal attacks. They guarantee the secure and expeditious transportation of patients to healthcare institutions for further treatment, while also conveying crucial information about the patient's state and the circumstances of their exposure to the medical team at the receiving end.

### Study duration

This study was completed in almost six months (16 August 2021 to 15 February 2022) which included the literature review, data collection, thesis writing, and data analysis.

### Sample size

All the emergency health care staff are included in the tertiary care hospitals of Peshawar. However, the total sample size was 100, and the distribution was as under [19]:

- Hayatabad Medical Complex: 30

- Lady Reading Hospital = 40

- Khyber Teaching Hospital = 30

### Sampling technique

Census included all the emergency health care staff in the study settings. Moreover, a convenient sampling technique was used for sample collection.

### Questionnaire

A standard questionnaire was used in the current study and was filled out by each participant after the verbal informed consent. The structured questionnaire was derived from previous KAP surveys done worldwide on rabies [13,20,21].

### Inclusion and Exclusion Criteria

All emergency staff willing to participate in the study were included. Housekeeping staff and those not involved in direct patient care (supervisors & in charge) were excluded from the study.

## Data collection procedure

After the approval from the graduate committee, ASRB, and ethical board, the researcher took permission from the administration of tertiary care hospitals to collect data from emergency health care staff regarding animal-bite-related rabies knowledge and practices. Written informed consent was taken from the study participants before data collection. The data on the knowledge and practice of emergency health care staff was collected via a validated questionnaire [19]. The Questionnaire used was in English and comprised of four sections.

1). Demographic 2). Knowledge 3). Attitude 4). Practices.

## Data analysis procedure

The data collected was analysed using SPSS version 22 and Microsoft Excel. Frequencies and percentages were calculated for categorical variables, e.g. gender, educational level, and category of healthcare workers (doctors, nurses, and paramedics staff). The participants with missing data were not entered into the Excel sheet.

## 3 Results

### Demographic characteristics of the participants

The study included a total of 68 males (68%) as shown in Table 1. The age-wise distribution revealed that majority of participants were of the younger age category (18–25 years, 44%). Profession-wise, nurses comprise the most prominent group (31%) compared to other professions, lag by medical officers (27%), house officers (26%), paramedical (13%), and faculty staff (3%) as shown in Table 1. The findings indicated that a significant proportion of participants had a very short duration of work experience. Relatively a small number of professionals have undergone training (to handle rabies-infected individuals according to WHO guidelines) in the treatment of rabies.

**Table 1. Descriptive statistics of respondents' demographic characteristics.**

| Variables | Response | Frequency | percentage |
|---|---|---|---|
| Gender | Males | 68 | 68% |
| | Females | 32 | 32% |
| Age-wise distribution of respondents | 18–25 | 44 | 44% |
| | 26–30 | 36 | 36% |
| | 31–35 | 16 | 16% |
| | 36–45 | 2 | 2% |
| | >45 | 2 | 2% |
| Professions of the respondents | Nurses | 31 | 31% |
| | Medical Officers | 27 | 27% |
| | House Officers | 26 | 26% |
| | Paramedics | 13 | 13% |
| | Faculty staff | 3 | 3% |
| Work experience | 0–5 Years | 77 | 77% |
| | 6–10 Years | 18 | 18% |
| | 11–15 Years | 2 | 2% |
| | >15 Years | 3 | 3% |
| Training | Yes | 18 | 18% |
| | No | 82 | 82% |

## Attitude and Practices of Respondents

Table 2 examined the attitudes and behaviours of healthcare personnel about rabies in different tertiary care hospitals located in Peshawar, Pakistan. A total of 91% of professionals accurately identified dogs as the principal carriers of rabies while a mere 9% acknowledged the possibility for cats to transmit rabies. Alarmingly, 16% of individuals rejected the idea of transfer between humans. 82% of professionals possessed an accurate understanding of transmission via animal bites, whilst 16% lacked awareness. Merely 31% recognised the possible spread via contaminated food or water as shown in Table 2. 92% of professionals possessed knowledge about the symptoms and prognosis of rabies, and 88% expressed a belief in saving those who have been bitten by rabid animals as shown in Table 2.

## Relationship between healthcare professionals work type and knowledge

Table 3 examined the relation between the specialisations of healthcare professionals and their views about rabies knowledge. Regarding the animals involved in spreading rabies, medical officers and nurses show a greater level of knowledge, and about 27% of medical officers and 24% of nurses accurately identified dogs as the main carriers of the disease. Conversely, house officers and paramedics demonstrated less knowledge, as just 2% of house officers and 13% of paramedics recognised dogs as carriers of rabies. A substantial statistical relation (p = 0.024) suggested that the healthcare profession's specialisation had influenced their knowledge of the primary carriers of rabies. Regarding the spread of rabies across individuals, healthcare professionals exhibited variability. Nurses demonstrate a greater percentage (27%) of recognising this possibility, while house officers indicated a lower percentage (14%). On the other hand, medical officers and faculty staff exhibited lesser levels of understanding in this regard (p = 0.007). Regarding comprehending how rabies is transmitted by animal bites, licks, or scratches, the responsibilities of healthcare professionals do not have a substantial impact on their level of knowledge, except for paramedics, who exhibited a slightly higher degree of doubt (p = 0.051).

There is a significant disparity in the comprehension of how rabies is transmitted by contaminated water (p = 0.002), with medical Officers and faculty staff demonstrating more awareness compared to house officers and paramedics. The results indicated a statistically significant preference for saving humans who are attacked by rabid animals among medical officers, house officers, and nurses (p = 0.040). Nevertheless, medical officers and house officers exhibited more doubt about the efficacy of home remedies for bite wounds (p = 0.033) as shown in Table 3.

## 4 Discussion

Rabies remains a significant global public health problem, particularly in developing countries such as Pakistan. Since rabies is regarded as a neglected tropical disease, limited public knowledge and awareness campaigns are conducted across the globe. Recent work from WHO under the umbrella of "Zero Rabies by 2030" has resulted in many countries' efforts to minimise the risk of rabies due to animal bites [20,22,23]. The present study showed that most respondents were aware that dogs and cats can spread rabies and that the disease can be transmitted via bites or licks from rabid animals. This finding was consistent with prior KAP surveys in India, Ethiopia, and Grenada [24].

The findings of this study are consistent with other studies conducted by Hampson et al. (2008) [5,25] and Sudarshan et al. (2006) [26,27], which indicate that a significant proportion of healthcare professionals acknowledge dogs as the principal transmitters of rabies. The diminished acknowledgment of felines as carriers aligns with research undertaken in Pakistan

**Table 2. Attitude and Practice of the respondents to rabies.**

| Parameters | Respondents | Frequencies | percentages |
|---|---|---|---|
| 1. Knowledge about responsible animals in most cases. | Dogs | 91.00 | 91% |
|  | Cats | 9.00 | 9% |
| 2. Knowledge about the human-to-human spread. | Yes | 76.00 | 76% |
|  | Not | 16.00 | 16% |
|  | Not Sure | 8.00 | 8% |
| 3. Knowledge about the spread of rabies via an animal bite | Yes | 82.00 | 82% |
|  | Not | 16.00 | 16% |
|  | Not Sure | 2.00 | 2% |
| 4. Knowledge about spreading of rabies via licks or scratches of an animal | Yes | 73.00 | 73% |
|  | No | 20.00 | 20% |
|  | Not Sure | 7.00 | 7% |
| 5. Knowledge regarding the spread of rabies through contaminated food\water. | Yes | 31.00 | 31% |
|  | No | 58.00 | 58% |
|  | Not Sure | 11.00 | 11% |
| 6. Knowledge regarding the appearance of symptoms and prognosis of rabies | Appear | 92.00 | 92% |
|  | Not sure | 8.00 | 8% |
| 7. Knowledge about the saving of a person bitten by a rabid animal with rabies | Yes | 88.00 | 88% |
|  | No | 8.00 | 8% |
|  | Not Sure | 4.00 | 4% |
| 8. Knowledge regarding the most typical treatment given to an animal bitten person | Apply local treatment | 2.00 | 2% |
|  | Apply antiseptic cream | 9.00 | 9% |
|  | Wash with Soap\Detergents | 44.00 | 44% |
|  | Apply antibiotics | 24.00 | 24% |
|  | Others | 21.00 | 21% |
| 9. Perception about treating an animal bite wound with home remedies | Yes | 30.00 | 30% |
|  | No | 57.00 | 57% |
|  | Not Sure | 13.00 | 13% |
| 10. Perception about washing an animal bite wound Soap/Detergents | Yes | 70.00 | 70% |
|  | No | 13.00 | 13% |
|  | Not Sure | 17.00 | 17% |
| 11. Knowledge about the duration of washing an animal bite wound with soap/detergent | All Reply 6–10 minutes | 100.00 | 100% |
| 12. Practice of suturing an animal bite wound at study hospitals | Yes | 24.00 | 24% |
|  | No | 39.00 | 39% |
|  | Some Time | 37.00 | 37% |
| 13. Perception about the observation of animal who had bitten the patient | Yes | 79.00 | 79% |
|  | No | 16.00 | 16% |
|  | Not Sure | 6.00 | 6% |
| 14. Duration of observation of animal who had bitten the patient | Ten days | 73.00 | 73% |
|  | 15 days | 13.00 | 13% |
|  | 20 days | 14.00 | 14% |
| 15. Knowledge regarding the post-exposure prophylactic treatment of rabies | Have knowledge | 81.00 | 81% |
|  | No knowledge | 14.00 | 14% |
|  | Not sure | 5.00 | 5% |
| 16. Type of post-exposure prophylactic treatment of rabies in the study hospital | ARV | 82.00 | 82% |
|  | RIG | 18.00 | 18% |

(*Continued*)

**Table 2.** (Continued)

| Parameters | Respondents | Frequencies | percentages |
|---|---|---|---|
| 17. Knowledge regarding availability of anti-rabies vaccine (ARV) in the vicinity of study hospitals | Yes | 78.00 | 78% |
| | No | 21.00 | 21% |
| | No idea | 1.00 | 1% |
| 18. Knowledge regarding the availability of rabies immunoglobulin (RIG) in the vicinity of the hospital | Yes | 60.00 | 60% |
| | No | 33.00 | 33% |
| | Not sure | 7.00 | 7% |
| 19. Practice of ARV | 5 doses | 89.00 | 89% |
| | 3 doses | 11.00 | 11% |
| 20. Perception about the administration of RIG directly after bite | Directly administered | 76.00 | 76% |
| | Not directly | 17.00 | 17% |
| | Not sure | 7.00 | 7% |
| 21. Perception about the administration of RIG up to 7 days of animal bite | Up to 7 days | 66.00 | 66% |
| | Not up to 7 days | 21.00 | 21% |
| | No Idea | 13.00 | 13% |
| 22. Practice of administration of ARV to a pregnant or lactating woman | Yes | 78.0 | 78% |
| | No | 15.00 | 15% |
| | No Idea | 7.00 | 7% |
| 23. Practice of administration of ARV to the patient even in case of unavailability of suspected animal | Yes | 75.00 | 75% |
| | No | 21.00 | 21% |
| | No idea | 4.00 | 4% |
| 24. Perception about the essential factor for failure of controlling rabies in human | Unavailability of ARV/RIG/PEP | 41.00 | 41% |
| | Lack of awareness | 20.00 | 20% |
| | Lack of control of stray dogs | 34.00 | 34% |

[28]. The recognition of the possibility of human-to-human transmission in your research aligns with the acknowledgment that, while seldom, such transmission is possible [29].

The study indicates that Medical Officers and Nurses tend to have elevated levels of awareness, which aligns with previous research conducted in India [27]. This variety underscores the need to implement focused training and treatments customized to healthcare professionals' distinct responsibilities. The treatment procedures found in the research are consistent with the results reported in Ethiopia by Abera et al. (2014) [30].

The results presented in this study provide insights into the correlation between the speciality of healthcare professionals and their knowledge and attitude pertaining to rabies. These findings are crucial for improving the understanding and implementation of rabies awareness and control strategies within hospital environments. The present discourse critically analyses the findings, drawing comparisons with other research works and proposing potential implications for future investigations and public health endeavors.

The observed correlation between healthcare professions and knowledge of dogs as the principal transmitters of rabies is in accordance with prior research findings [9,31]. Medical officers and nurses, who serve as primary healthcare professionals, have a heightened level of consciousness, underscoring the significance of their contributions in the realm of rabies education and prevention. The observed disparity in understanding about the transfer of the human-to-human deserves attention. Nurses and house officers, who often engage in direct patient care, have heightened levels of awareness. The discovery highlights the capacity of individuals to serve as instructors in the field of rabies prevention [32]. According to Sudarshan et al. (2019) [33,34], there is little evidence to imply that healthcare positions substantially

**Table 3. Association between work type and knowledge perspective of the respondent about the rabies.**

| Factor | Detail | Medical Officer | House Officer | Faculty Staff | Nurse | Paramedics | P-value |
|---|---|---|---|---|---|---|---|
| Animal Responsible | Dog | 27 (27%) | 24 (24%) | 3 (3%) | 24 (24%) | 13 (13%) | 0.024 |
| | Cat | 0 (0%) | 2 (2%) | 0 (0%) | 7 (7%) | 0 (0%) | |
| Human to Human Transmission | Yes | 23 (23%) | 14 (14%) | 1 (1%) | 27 (27%) | 11 (11%) | 0.007 |
| | No | 4 (4%) | 6 (6%) | 2 (2%) | 3 (3%) | 1 (1%) | |
| | Not Sure | 0 (0%) | 6 (6%) | 0 (0%) | 1 (1%) | 1 (1%) | |
| Rabies can spread through bite | Yes | 21 (21%) | 20 (20%) | 3 (3%) | 25 (25%) | 13 (13%) | 0.306 |
| | No | 6 (6%) | 6 (6%) | 0 (0%) | 4 (4%) | 0 (0%) | |
| | Not Sure | 0 (0%) | 0 (0%) | 0 (0%) | 2 (2%) | 0 (0%) | |
| Rabies can spread through licks scratches | Yes | 22 (22%) | 20 (20%) | 1 (1%) | 20 (20%) | 10 (10%) | 0.051 |
| | No | 4 (4%) | 6 (6%) | 2 (2%) | 5 (5%) | 3 (3%) | |
| | Not Sure | 1 (1%) | 0 (0%) | 0 (0%) | 6 (6%) | 0 (0%) | |
| Rabies can spread through contaminated water | Yes | 13 (13%) | 3 (3%) | 2 (2%) | 11 (11%) | 2 (2%) | 0.002 |
| | No | 13 (13%) | 21 (21%) | 1 (1%) | 12 (12%) | 11 (11%) | |
| | Not sure | 1 (1%) | 2 (2%) | 0 (0%) | 8 (8%) | 0 (0%) | |
| The person dies if bitten by rabid animal | Yes | 24 (24%) | 26 (26%) | 3 (3%) | 26 (26%) | 13 (13%) | 0.147 |
| | No | 3 (3%) | 0 (0%) | 0 (0%) | 5 (5%) | 0 (0%) | |
| Can be saved if bitten? | Yes | 23 (23%) | 26 (26%) | 3 (3%) | 23 (23%) | 13 (13%) | 0.040 |
| | No | 4 (4%) | 0 (0%) | 0 (0%) | 4 (4%) | 0 (0%) | |
| | Not Sure | 0 (0%) | 0 (0%) | 0 (0%) | 4 (4%) | 0 (0%) | |
| Is treatment with home remedies useful? | Yes | 10 (10%) | 5 (5%) | 0 (0%) | 10 (10%) | 5 (5%) | 0.033 |
| | No | 17 (17%) | 18 (18%) | 3 (3%) | 12 (12%) | 7 (7%) | |
| | Not Sure | 0 (0%) | 3 (3%) | 0 (0%) | 9 (9%) | 1 (1%) | |
| Is washing the bitten site useful as a treatment purpose? | Yes | 23 (23%) | 19 (19%) | 1 (1%) | 17 (17%) | 10 (10%) | 0.010 |
| | No | 2 (2%) | 2 (2%) | 2 (2%) | 4 (4%) | 3 (3%) | |
| | Not Sure | 2 (2%) | 5 (5%) | 0 (0%) | 10 (10%) | 0 (0%) | |
| Is suturing the dog bitten site useful? | Yes | 6 (6%) | 1 (1%) | 2 (2%) | 11 (11%) | 4 (4%) | 0.041 |
| | No | 9 (9%) | 10 (10%) | 1 (1%) | 13 (13%) | 6 (6%) | |
| | Not Sure | 12 (12%) | 15 (15%) | 0 (0%) | 7 (7%) | 3 (3%) | |
| Does treatment prevent the rabies patients? | Yes | 27 (27%) | 24 (24%) | 3 (3%) | 17 (17%) | 10 (10%) | 0.003 |
| | No | 0 (0%) | 2 (2%) | 0 (0%) | 10 (10%) | 2 (2%) | |
| | Not Sure | 0 (0%) | 0 (0%) | 0 (0%) | 4 (4%) | 1 (1%) | |
| ARV readily available | Yes | 26 (26%) | 14 (14%) | 3 (3%) | 27 (27%) | 8 (8%) | 0.002 |
| | No | 1 (1%) | 12 (12%) | 0 (0%) | 4 (4%) | 4 (4%) | |
| | Not Sure | 0 (0%) | 0 (0%) | 0 (0%) | 0 (0%) | 1 (1%) | |

*(Continued)*

**Table 3.** (Continued)

| Factor | Detail | Medical Officer | House Officer | Faculty Staff | Nurse | Paramedics | P-value |
|---|---|---|---|---|---|---|---|
| Is any schedule available in your hospital? | Yes | 23 (23%) | 25 (25%) | 3 (3%) | 21 (21%) | 7 (7%) | 0.013 |
| | No | 3 (3%) | 1 (1%) | 0 (0%) | 4 (4%) | 5 (5%) | |
| | Not Sure | 1 (1%) | 0 (0%) | 0 (0%) | 6 (6%) | 1 (1%) | |
| Is RIG scheduled after 7 days? | Yes | 16 (16%) | 21 (21%) | 3 (3%) | 14 (14%) | 12 (12%) | 0.002 |
| | No | 10 (10%) | 3 (3%) | 0 (0%) | 7 (7%) | 1 (1%) | |
| | Not Sure | 1 (1%) | 2 (2%) | 0 (0%) | 10 (10%) | 0 (0%) | |
| Is rabies treatable after one month? | Yes | 24 (24%) | 18 (18%) | 1 (1%) | 20 (20%) | 7 (7%) | 0.032 |
| | No | 3 (3%) | 8 (8%) | 2 (2%) | 6 (6%) | 4 (4%) | |
| | Not Sure | 0 (0%) | 0 (0%) | 0 (0%) | 5 (5%) | 2 (2%) | |
| Factors involved in failure of control on rabies infection? | Non-availability of ARV/RIG/PEP | 6 (6%) | 17 (17%) | 0 (0%) | 12 (12%) | 6 (6%) | 0.009 |
| | Lack of Awareness among people | 4 (4%) | 5 (5%) | 0 (0%) | 8 (8%) | 3 (3%) | |
| | Lack of control over stray dogs | 15 (15%) | 3 (3%) | 3 (3%) | 11 (11%) | 2 (2%) | |
| | Others | 2 (2%) | 1 (1%) | 0 (0%) | 0 (0%) | 2 (2%) | |

impact understanding the transmission modes for rabies. However, it is worth noting that Paramedics exhibit a rather high level of ambiguity in this area. This finding highlights the need of implementing focused training programs that specifically address these knowledge gaps.

The observed disparity in understanding the transmission of rabies via water that has been polluted is in line with prior investigations conducted by Shaikh et al., (2023) [35,36]. Implementing awareness programs focused on this transmission mechanism can potentially boost the general understanding of House Officers and Paramedics. The presence of skepticism among Medical Officers and House Officers about using home medicines to treat bite wounds highlights the need to establish clear guidelines and best practices in this area [37,38]. The endorsement of rescuing those affected by rabies is commendable and in accordance with international efforts to advocate for prompt medical intervention [39].

The notion of the availability of antiretroviral (ARV) treatment holds promise, yet there are notable gaps that need to be addressed. It is imperative to prioritize effective supply chain management to ensure consistent access to ARVs [33,36]. The divergence in perspectives regarding rabies immunoglobulin (RIG) scheduling may necessitate additional training and guidance. The factors identified in the failure to control rabies align with findings from previous research studies [37,38]. Taking measures to address the unavailability of post-exposure prophylaxis and raising awareness are critical steps in controlling rabies.

## Benefits

This study presents several significant advantages, such as augmenting healthcare workers' knowledge and understanding of rabies and improving precision and efficacy in patient treatment. This study provides valuable insights into the many elements that influence the control of rabies and the accessibility of post-exposure prophylaxis, hence possibly playing a crucial

role in safeguarding public health and perhaps preventing loss of life. This research facilitates the development of customized training programs for healthcare workers, enhancing their effectiveness in the realm of rabies prevention and control. Policymakers may use this study's results to develop policies grounded on empirical data, guaranteeing that healthcare facilities have the necessary resources and capabilities to manage cases of rabies effectively. Moreover, this study establishes a fundamental basis for forthcoming investigations in the discipline, offering significant perspectives with worldwide health ramifications. In conclusion, the article's influence encompasses lowering costs and the potential to enhance public health outcomes by improving the efficiency of rabies preventive and management strategies.

The educational background and duration of study have a substantial impact on healthcare professionals' understanding and application of rabies prevention and treatment techniques. Individuals with significant medical expertise, such as nurses and medical officers, may have a higher level of awareness compared to paramedics and house officers. Hence, our research recognizes the need of clearly define these characteristics of comparability to improve the understandability and applicability of our results.

Our research acknowledges the need to further investigate these consequences to provide valuable insights for the development of focused educational efforts and training programs that seek to enhance the accessibility and effectiveness of therapy. By clarifying these consequences, we contribute to the advancement of data-driven methods for improving rabies management techniques and eventually benefiting public health outcomes.

## Limitations of the study

The data depends on self-reported information provided by healthcare providers, which introduces the possibility of recollection bias or social desirability bias, possibly impacting the replies' precision. Additionally, it should be noted that although the sample size provides valuable insights, it may not adequately capture the complete range of healthcare practitioners within the area. Furthermore, it should be noted that the research in question is cross-sectional, which inherently restricts its capacity to demonstrate a causal relationship. In conclusion, the study offers significant contributions to the understanding of healthcare professionals' knowledge and practices in the context of rabies. However, it is important to note that the research did not extensively explore the efficacy of targeted treatments designed to enhance rabies awareness and control. Further research is necessary to investigate future studies that include bigger and more varied populations, longitudinal designs, and intervention evaluations.

## Future perspectives

In subsequent investigations, it is essential to prioritize the implementation and assessment of specific interventions to enhance the knowledge and control of rabies among healthcare personnel. Longitudinal studies have the capacity to monitor and evaluate the efficacy of training programs by examining their long-term influence on knowledge acquisition, attitudes, and behavioral patterns. Furthermore, it is essential to investigate cooperation endeavors between the veterinary and public health sectors to advance the One Health strategy, which aims to tackle the issue of rabies. Further investigation is required to assess the cost-effectiveness of interventions and the long-term viability of healthcare systems in providing services associated with rabies. In conclusion, the establishment of global partnerships has the potential to enable the sharing of information, promoting a collaborative endeavor aimed at reducing the incidence of rabies and enhancing patient care worldwide.

## 5 Conclusion

In summary, this research offers significant contributions to understanding the knowledge, attitudes, and behaviors shown by healthcare personnel in relation to rabies inside different tertiary care hospitals located in Peshawar, Pakistan. The study's results indicate that there are diverse degrees of knowledge and perspectives among individuals in various healthcare jobs. This highlights the need to implement educational interventions that are specifically designed to meet the needs of each function. Medical Officers and Nurses often have a greater level of understanding, while House Officers and Paramedics show deficiencies in some areas of expertise. The findings of this study underscore the need to implement role-specific training programs to address these inequities and improve the effectiveness of rabies prevention and management efforts. Additionally, the research examines the many variables that impact rabies control, emphasizing the need to address the lack of access to post-exposure prophylaxis and increasing public awareness. This discovery makes a significant contribution to the wider public health endeavor aimed at addressing rabies, with the potential to save lives and alleviate the impact of this highly lethal disease. Future endeavors should prioritize implementing and assessing specific interventions and promoting collaborative methods rooted in the One Health framework to prevent the spread of rabies effectively.

## Supporting information

**S1 Data. The file named "S1 data" uploaded in the supporting information is an SPSS file containing the raw data for the entire manuscript.**
(RAR)

## Author Contributions

**Conceptualization:** Fawad Inayat, Shaista Rasul.

**Data curation:** Adnan Ahmad, Shehnaz Bakhtiar, Zar Shad.

**Formal analysis:** Adnan Ahmad, Fawad Inayat, Zakir Ahmad.

**Investigation:** Adnan Ahmad, Fawad Inayat, Noor Ullah.

**Methodology:** Fawad Inayat, Noor Ullah, Shaista Rasul.

**Project administration:** Adnan Ahmad, Fawad Inayat, Noor Ullah, Shaista Rasul.

**Resources:** Adnan Ahmad, Shaista Rasul.

**Software:** Fawad Inayat, Zakir Ahmad.

**Supervision:** Noor Ullah, Shaista Rasul, Shehnaz Bakhtiar.

**Validation:** Noor Ullah, Shaista Rasul.

**Visualization:** Adnan Ahmad, Fawad Inayat, Zar Shad.

**Writing – original draft:** Adnan Ahmad, Fawad Inayat.

**Writing – review & editing:** Fawad Inayat, Noor Ullah, Shaista Rasul, Shehnaz Bakhtiar, Zar Shad, Zakir Ahmad.

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
