## [Decision Letter · Decision Letter 0]

13 Dec 2023

Dear Dr Bakhtiar,

Thank you very much for submitting your manuscript "Knowledge, Attitudes, and Practices of Healthcare Professionals Regarding Rabies in Tertiary Care Hospitals: A Cross-Sectional Study in Peshawar, Pakistan" for consideration at PLOS Neglected Tropical Diseases. As with all papers reviewed by the journal, your manuscript was reviewed by members of the editorial board and by several independent reviewers. In light of the reviews (below this email), we would like to invite the resubmission of a significantly-revised version that takes into account the reviewers' comments. 

We cannot make any decision about publication until we have seen the revised manuscript and your response to the reviewers' comments. Your revised manuscript is also likely to be sent to reviewers for further evaluation.

Sincerely,

Marilia Sá Carvalho

Academic Editor

Michael Holbrook

Section Editor

Reviewer's Responses to Questions

**Key Review Criteria Required for Acceptance?**

**Methods**

-Are the objectives of the study clearly articulated with a clear testable hypothesis stated?

-Is the study design appropriate to address the stated objectives?

-Is the population clearly described and appropriate for the hypothesis being tested?

-Is the sample size sufficient to ensure adequate power to address the hypothesis being tested?

-Were correct statistical analysis used to support conclusions?

-Are there concerns about ethical or regulatory requirements being met?

Reviewer #1: Title and summary are concise and clear.

Line 36 - It was not clear to which places the authors are referrring regarding the incidence rate if rabies-related deaths:

India - 20 to 30 cases per million per year.

Not described (Bangladesh?) - 14 cases per million per year.

Pakistan - 7 to 9 cases per million per year

Methodology – It was not described which health professionals were included in the sample.

The described methodology supports the achievement of the proposed objectives 

The research was conducted after approval by the ethics committe and with the consente of healthcare professionals in hospitals in Pakistan.

Reviewer #2: The article “Knowledge, Attitudes, and Practices of Healthcare Professionals Regarding Rabies in

Tertiary Care Hospitals: A Cross-Sectional Study in Peshawar, Pakistan” addresses an important topic but I have following concerns,

1. The introduction doesn’t provide sufficient background in the topic of research and the objectives are not clearly defined. 

2. The study duration is missing only a period of 6 months has been mentioned.

3. The authors have not mentioned how they have collected the sample size?

4. The ethical approval number is missing.

5. No details are provided regarding the questionnaire used for the study and what was the language of the questionnaire and how it was validated?

Reviewer #3: I have carefully reviewed the Methods section of your research manuscript. Overall, the study design appears to be robust, but there are some discrepancies and areas that require further clarification. Below is a detailed analysis of the comments provided:

Design Clarification:

The term "design" should be explicitly defined in the Methods section. Please provide a clear and detailed explanation of the study design employed.

Separation of Questionnaire and Consent:

It is recommended to remove the questionnaire and consent from the main text in study design and present them separately. This enhances clarity and allows readers to focus on the research methods without distraction.

Duration of the Study:

The duration of the study should be precisely stated in terms of start and end dates. Remove unnecessary wordings and provide an exact time frame for the study period.

Hospital Selection and Sample Size Calculation:

Clarify the criteria used for selecting the participating hospitals. Additionally, elaborate on how the sample size was determined, addressing issues related to statistical power and whether the study meets the required power for significance testing.

Languages Used in Questionnaire:

Specify the languages in which the questionnaire was conducted. Additionally, explain the process of translating questions from other studies into the required languages. This information is crucial for understanding the linguistic validity of the instrument.

Sampling Technique and Sample Selection:

Explicitly state the sampling technique employed in the study. Provide details on how participants were selected, ensuring transparency in the sampling process.

Ethical Board Approval and Questionnaire Reference:

Include the ethical board approval number and reference for the questionnaires used in your study. This adds credibility to your research and ensures compliance with ethical standards.

Survey Language:

Clearly state whether the survey was conducted in English, Urdu, or both. Additionally, provide more details on the items and sections of the questionnaire to give readers a comprehensive understanding of the data collection process.

Data Analysis Description:

Elaborate on the data analysis methods used in detail. Include any specific procedures for handling missing data and statistical test used. This will enhance the transparency and reproducibility of your findings.

**Results**

-Does the analysis presented match the analysis plan?

-Are the results clearly and completely presented?

-Are the figures (Tables, Images) of sufficient quality for clarity?

Reviewer #1: The acronym (ARV) in table 2 is wrong:

Knowledge regarding availability of anti-rabies vaccine (AVR) in the vicinity of study hospitals

In table 3, the values related to the variable “Rabies can spread through licks scratches” are wrong.

Reviewer #3: I have carefully reviewed the manuscript, and I would like to provide the following comments for your consideration:

Deletion of First Two Lines (154, 155):

I suggest removing the first two lines (154, 155) as they appear redundant or unnecessary in the context of the manuscript.

Revision of Sentences (157-159):

The sentences in this section give the impression that the survey is conducted in a single hospital. Please consider revising these sentences to accurately convey the scope and setting of the survey to avoid potential misinterpretations.

Deletion of a Sentence (Regarding Gender Inequalities):

I recommend removing the sentence that discusses possible ramifications for the findings of the research due to pre-existing gender inequalities in the healthcare sector. This statement appears to be out of context and doesn't contribute to the clarity of the manuscript.

Overall Results Revision:

The overall results section needs a comprehensive revision for improved clarity. Consider reorganizing and rewriting this section from scratch to enhance the readability and coherence of the presented information. Providing a clear and concise summary of the research findings will significantly strengthen the manuscript.

Ambiguity in the Term "Training" (Table 1):

The term "Training" in Table 1 is deemed ambiguous. I recommend clearly defining or specifying the type of training being referred to in the table to eliminate any potential confusion for readers. Additionally, please ensure the table is formatted more clearly, and consider removing the percentage sign for better presentation.

Results Section Revamp:

The results section, as it stands, was challenging to follow. A complete revision is necessary to enhance the clarity and coherence of the information presented. Provide a concise and organized summary of the research findings, ensuring that the narrative is logically structured and easy to comprehend.

**Conclusions**

-Are the conclusions supported by the data presented?

-Are the limitations of analysis clearly described?

-Do the authors discuss how these data can be helpful to advance our understanding of the topic under study?

-Is public health relevance addressed?

Reviewer #1: The conclusions are supported by the data presented, corroborating similar studies in other locations with similar contexts.

Reviewer #3: NA

**Editorial and Data Presentation Modifications?**

Reviewer #1: Minor Revision

Reviewer #2: (No Response)

Reviewer #3: Before sending the manuscript for peer review, the editorial desk should conduct a careful check of its worth and scope, language and structure, as well as its conformity to the journal guidelines.

**Summary and General Comments**

Reviewer #1: Rabies is a disease that is essentially 100% lethal, yet 100% preventable through the implementation of appropriate and timely control measures. This study aimed to assess the knowledge, attitudes, and practices of healthcare professionals working in the emergency sector of three tertiary hospitals. The survey results hold significance as they provide insights into the rabies knowledge, attitudes, and practices of emergency professionals in Pakistan, a region with a high rabies mortality rate.

It would be valuable to briefly discuss the variance in training among different professional categories, as the information they possess may vary based on their level of education and expertise.

The authors might consider exploring the significance of each professional's role in providing care to individuals at risk of rabies. Among the professionals included in the sample, who plays the most crucial role in prescribing prophylaxis through vaccination, immunoglobulin administration, wound care, treatment, and the notification of suspected human rabies cases?

We recommend including an overview of these hospitals' impact on the local healthcare network. Are these hospitals primarily responsible for rabies care?

It would be enlightening to understand what proportion of hospitals provide similar care and whether these hospitals serve as reference points for individuals exposed to the rabies virus. This information is crucial for establishing an integrated action plan for continuous and ongoing education with a unified healthcare perspective.

Reviewer #2: (No Response)

Reviewer #3: Abstract:

Background: Rather than describing rabies as a significant global public health issue, characterize it in terms of endemicity, specifying the number of affected countries, considering that some are rabies-free. Use abbreviations consistently, such as Health Care Professional (HCP).

As there is no definitive treatment, replace the terms "prevention and treatment" with "control and prevention" or "preventive treatment" in the wording. Revise the objective for clarity, avoiding the use of synonyms.

In methods, address inconsistencies and eliminate redundancy. The method lacks representativeness, with unnecessary wording used to extend the text. For qualitative variables, use "association" instead of "correlation" in the context of describing the correlation.

Maintain harmony: either use "healthcare professionals" or "healthcare workers." Please observe the use of uppercase letters.

In results, state that males were in preponderance, providing numbers and percentages. Justify the inclusion of faculty staff in the method section. Avoid starting sentences with numbers and use the full form for standard abbreviations and acronyms in subsequent text.

Regarding "significant disparities in roles," the given text is ambiguous, and clarification is needed on what roles the author is addressing. Be consistent with the denotation of numbers (%) throughout the text.

In conclusion, base recommendations on the study findings rather than on subjects that haven't been discussed.

Introduction:

Please focus on describing the virus itself rather than its manifestations.

Line 81: The statistics related to Asians and Africans cannot be supported by a single reference.

The overall text predominantly centers around the incidence and prevalence. Although KAP results have been elucidated in relation to other nations, there is an abundance of published studies on the subject from this particular region. Furthermore, the transition to the implications of the study appears abrupt, lacking a thorough exploration of the rationale and existing gaps. It is imperative to provide a robust rationale for the study.

Methods:

The Methods section requires meticulous drafting. The term "design" should be clarified for precision.

Separate the questionnaire and consent, omitting redundancy.

Condense the study's duration by eliminating unnecessary wording and specify the exact timeframe.

Rationalize the selection of hospitals and clarify if the sample size was calculated.

Address the potential underpowered state of the study; with only 100 responses, confirm whether statistical power requirements were met or if the findings indicate a trend without statistical significance. If the study achieved the required power, explicitly mention it.

Specify the languages in which the questionnaire was administered and elaborate on the translation process for incorporating questions from other studies into the desired languages.

Clearly articulate the sampling technique and delineate the criteria for sample selection.

Include the ethical board approval number and reference for the questionnaires.

Specify whether the survey was conducted in English, Urdu, or both, and provide additional details on the questionnaire items and sections.

Elaborate on the data analysis process, providing comprehensive details.

Methods:

Results: Please remove the first two lines (154, 155). 

Please revise sentences 157-159, as they create the impression that the survey is conducted in a single hospital.

I suggest deleting the sentence: "which may have ramifications for the findings of the research, presumably reflecting pre-existing gender inequalities in the healthcare sector." There is no need for possible explanations of out-of-

---

## [Decision Letter · Decision Letter 1]

3 Apr 2024

Dear Dr Bakhtiar,

Thank you very much for submitting your manuscript "Knowledge, Attitudes, and Practices of Healthcare Professionals Regarding Rabies in Tertiary Care Hospitals: A Cross-Sectional Study in Peshawar, Pakistan" for consideration at PLOS Neglected Tropical Diseases. As with all papers reviewed by the journal, your manuscript was reviewed by members of the editorial board and by several independent reviewers. In light of the reviews (below this email), we would like to invite the resubmission of a significantly-revised version that takes into account the reviewers' comments. 

We cannot make any decision about publication until we have seen the revised manuscript and your response to the reviewers' comments. Your revised manuscript is also likely to be sent to reviewers for further evaluation.

Sincerely,

Marilia Sá Carvalho

Academic Editor

Michael Holbrook

Section Editor

Reviewer 1

Methods

The research describes the inclusion of 4 professional categories that are closely involved in patient care. However, considering the regional differences among these professional categories, it would be important to include, in a summarized manner, the role of these professionals in the stages of patient care for individuals who have had accidents involving potential rabies-transmitting animals and those patients showing signs and symptoms suspicious of the disease.

It would be interesting to include the criteria for selecting the three hospitals included in the sample. Are these hospitals reference centers for pre- and post-exposure prophylaxis and for treating individuals with rabies?

The "Data analysis procedure" section is in the future tense. It wasn't described which statistical methods were used for data analysis. Were only frequencies and percentages used?

Results

Results are presented, and some of them are discussed in this section. I suggest using this section solely to describe the main findings of the research.

The discussions in this section lead to suggestions that need to be reviewed considering only one variable. For example: Would it be possible to sustain an association between years of professional experience and lack of specific rabies training with potential deficiencies in readiness and understanding in describing the demographic characteristics of the participants?

Tables 1 and 2 have categories that do not follow the same standardization. I suggest presenting the categories in descending order. For example: Professions of the Respondents.

In the section "Association between respondents' work type and knowledge," I suggest replacing the word "association" with "relationship." No association measures were conducted.

Regarding the results in Table 3, it would be interesting to focus according to the professional category of the respondents. Considering the education, years of study, and roles performed by each professional category in caring for people exposed to the rabies virus, would there be different response profiles?

In Table 3, it would be interesting to include the count (n) and percentage (%) for each response.

I suggest including a discussion about the comparability among the professional categories included in the sample. Considering the level of education and years of study, are doctors and nurses comparable to paramedics and home officers?

As a suggestion, for example, regarding the question "Does treatment prevent rabies in patients?" If nurses do not know, could it signify barriers to access adequate treatment? Similarly, if a paramedic does not know, would it be indicative of a barrier to access?
---

## [Editor Report · Decision Letter 2]

22 May 2024

Dear Dr Bakhtiar,

We are pleased to inform you that your manuscript 'Knowledge, Attitudes, and Practices of Healthcare Professionals Regarding Rabies in Tertiary Care Hospitals: A Cross-Sectional Study in Peshawar, Pakistan' has been provisionally accepted for publication in PLOS Neglected Tropical Diseases.

Best regards,

Marilia Sá Carvalho

Academic Editor

Michael Holbrook

Section Editor

---

## [Editor Report · Acceptance letter]

3 Jun 2024

Dear Dr Bakhtiar,

We are delighted to inform you that your manuscript, "Knowledge, Attitudes, and Practices of Healthcare Professionals Regarding Rabies in Tertiary Care Hospitals: A Cross-Sectional Study in Peshawar, Pakistan," has been formally accepted for publication in PLOS Neglected Tropical Diseases.

Best regards,

Shaden Kamhawi

co-Editor-in-Chief

Paul Brindley

co-Editor-in-Chief
